# Computing Representations for Lie Algebraic Networks

**Noah Shutty**[*]                                         NOAHSHUTTY@GMAIL.COM  
**Casimir Wierzynski**[†]                                    CASIMIR@CASIMIR.NET  
*Intel AI, San Diego, CA*

**Editors:** Sophia Sanborn, Christian Shewmake, Simone Azeglio, Arianna Di Bernardo, Nina Miolane

## Abstract

Recent work has constructed neural networks that are equivariant to continuous symmetry groups such as 2D and 3D rotations. This is accomplished using explicit *Lie group representations* to derive the equivariant kernels and nonlinearities. We present three contributions motivated by frontier applications of equivariance beyond rotations and translations. First, we relax the requirement for explicit Lie group representations with a novel algorithm that finds representations of arbitrary Lie groups given only the *structure constants* of the associated Lie algebra. Second, we provide a self-contained method and software for building Lie group-equivariant neural networks using these representations. Third, we contribute a novel benchmark dataset for classifying objects from relativistic point clouds, and apply our methods to construct the first object-tracking model equivariant to the Poincaré group.

**Keywords:** Lie Groups, Group Representations, Equivariance, Object Tracking

## 1. Introduction

Some regression and classification tasks on continuous input data obey a *continuous symmetry* such as 2D rotations. An ML model is said to be *equivariant* to the symmetry if the model respects it (without training). Equivariant models have been generalized beyond 2D rotations to other symmetries such as 3D rotations. These generalizations are enabled by mathematical results about each set of symmetries. Specifically, explicit *group representation matrices* for each new symmetry group are required. For many important symmetries, formulae are readily available to produce these representations. For other symmetries we are not so lucky: the representations may be difficult to compute explicitly and in the worst cases the classification of the group representations is an open question. For example, in the important case of the *homogeneous Galilean group*, which we define in section 2, the classification of the finite dimensional representations is a so-called "wild algebraic problem" for which we have only partial solutions (De Montigny et al., 2006; Niederle and Nikitin, 2006; Levy-Leblond, 1971).

    Novel approaches are needed to construct an equivariant network without prior knowledge of the group representations. In this work, we propose an algorithm **FindRep** that computes representation matrices with high numerical precision. We validate that FindRep succeeds for the *Poincaré group*, a set of symmetries governing tasks ranging from particle physics to object tracking, and on two other sets of symmetries where formulae are known. We apply the Poincaré group representations obtained by FindRep to construct

---

[*] Work done during an internship at Intel AI. Submitted while at the Department of Physics, Stanford University, Stanford, CA. Now at Google Quantum AI, Venice, CA.

[†] Now at Rockley Photonics Inc., Pasadena, CA.

**SpacetimeNet**, a Poincaré-equivariant object-tracking model. We provide all source code in software library titled *Lie Algebraic Networks*[1]. As far as we are aware, FindRep is the first automated solver which can find explicit representation matrices for sets of symmetries which form *noncompact, noncommutative Lie groups*. Further, SpacetimeNet is the first object-tracking model with a rigorous guarantee of Poincaré group equivariance.

## 1.1. Group Representations and Equivariant Machine Learning

Group theory provides the mathematical framework for describing symmetries and building equivariant ML models. Informally, a symmetry group $G$ is a set of invertible transformations $\alpha, \beta \in G$ which can be composed together using a product operation $\alpha\beta$. We are interested in continuous symmetries for which $G$ is a *Lie group*. In prior constructions of Lie group-equivariant models, *group representations* are required. For a group $G$, an $n-$dimensional (real) group representation $\rho : G \to \mathbb{R}^{n \times n}$ is a mapping from each element $\alpha \in G$ to an $n \times n$-dimensional matrix $\rho(\alpha)$, such that for all pairs $\alpha, \beta \in G$, we have $\rho(\alpha)\rho(\beta) = \rho(\alpha\beta)$. Several approaches to implementing Lie group equivariant neural networks have been developed in the literature and we defer to Weiler et al. (2021) for a comprehensive review. Here, we consider the approach taken by Thomas et al. (2018); Anderson et al. (2019); Bogatskiy et al. (2020), in which steerable convolutions and nonlinearities are performed directly on a field of irreducible group representations (irreps), which we define in section 2.4. These works utilize existing analytic formulas derived for the matrices of these representations. However, these formulas are only available for specific Lie groups where the representation theory is well-understood. A more convenient approach for extending equivariance to novel Lie groups would utilize an automated computational technique to obtain the required representations. Our primary contribution is such a technique.

## 1.2. Contributions

We automate finding explicit group representation matrices of Lie groups using an algorithm called **FindRep**. FindRep poses an optimization problem defined by the *Lie algebra* associated with a Lie group, whose solutions are the representations of the algebra. A penalty term is used to prevent the formation of *trivial representations*. Gradient descent of the resulting loss function produces nontrivial representations upon convergence. We apply FindRep to three noncommutative Lie groups for which the finite-dimensional representations are well-understood, allowing us to verify that the representations produced are irreps by computing their *Clebsch-Gordan coefficients* and applying *Schur's Lemma*.

One of the Lie groups where FindRep performs well is the Lorentz group of special relativity. Prior work has applied Lorentz-equivariant models to particle physics. In section 2.2 we observe that the Lorentz group along with the larger Poincaré group also governs everyday object-tracking tasks. We construct a Poincaré-equivariant neural network architecture called **SpacetimeNet** and demonstrate that it can learn to solve a 3D object-tracking task subject to "motion equivariance," where the inputs are a time series of points in space. Our work also motivates further study of Lorentz equivariance for object tracking.

---

1. github.com/noajshu/learning_irreps

### 1.3. Organization

We summarize related work in section 1.4. We summarize all necessary background and terminology in section 2. We describe FindRep in section 3.1 and SpacetimeNet in section A.2. We present our experimental results on learning Lie group irreps with FindRep in section 4 and on the performance of our Poincaré-equivariant SpacetimeNet model on a 3D object tracking task in appendix A.3. We defer the results on SpacetimeNet to an appendix because we consider FindRep to be our primary contribution towards a general technical framework and toolset for building neural networks equivariant to arbitrary Lie groups.

### 1.4. Related Work

In this section we summarize the most closely-related prior work and explain the novelty of our contributions. We defer to section 2 for relevant technical definitions and to Weiler et al. (2021) for a comprehensive literature review.

Several authors have investigated automated means of identifying Lie group representations. First, Rao and Ruderman (1999) used gradient descent to find the Lie group generators, given many examples of data which had been transformed by the group. Applying their technique requires knowledge of how the group acts on a representation space. In our case we know the Lie algebra's structure constants but we do not know how to compute its representations. Tai et al. (2019) gave a closed-form solution for the *canonical coordinates* for Lie groups, but their formula only applies for Abelian one-parameter Lie groups, excluding $SO(3), SO(2,1)$, and $SO(3,1)$. Cohen and Welling (2014) devised a probabilistic model to learn representations of *compact, commutative* Lie groups from pairs of images related by group transformations. In the present work we demonstrate a new approach to handle *noncompact* and *noncommutative* groups such as $SO(3), SO(2,1)$, and $SO(3,1)$. Finally, computer algebra software such as the LiE package developed by Van Leeuwen et al. (1992) can automate certain representation-theoretic computations for completely reducible Lie groups, but this software is of limited use when considering novel Lie groups where the representation theory is not well understood.

Beginning with the success of (approximately) translation-equivariant CNNs introduced by LeCun et al. (1989) for image recognition, a line of work has extended equivariance to additional continuous symmetry groups. Most relevant are the architectures for groups $SE(2)$ (Worrall et al., 2017; Weiler and Cesa, 2019), $SE(3)$ (Weiler et al., 2018; Cohen et al., 2019; Kondor et al., 2018; Thomas et al., 2018; Cohen et al., 2018; Kondor, 2018; Gao et al., 2020; Anderson et al., 2019; Fuchs et al., 2020; Eismann et al., 2020), and the group of Galilean boosts (Zhu et al., 2019).

The work by Thomas et al. (2018); Kondor et al. (2018); Anderson et al. (2019); Bogatskiy et al. (2020) used Clebsch-Gordan coefficients in their equivariant neural networks. Weiler et al. (2018), generalized by Cohen et al. (2019) showed all equivariant linear maps are convolutions whose kernels satisfy some linear constraints. In our work we obtain Clebsch-Gordan coefficients from similar linear constraints (eq. (5)) and use them to show that the learned representations are irreducible. We also use them in SpacetimeNet. Griffiths and Griffiths (2005) provide an introductory exposition of Clebsch-Gordan coefficients and Gurarie (1992) provides a more general exposition.

One of the first constructions that addressed spatiotemporal symmetries of object tracking was by Zhu et al. (2019). They introduce *motion-equivariant* networks to handle linear optical flow of an observer moving at a fixed velocity. They use a canonical coordinate system in which optical flow manifests as a translation, as generalized by Tai et al. (2019). This trick allows them to apply translation-equivariant CNNs to produce Galilean boost-equivariance, at the cost of giving up equivariance to translations of the original coordinate system. To maintain approximate translation-equivariance, the authors apply a spatial transformer network (Jaderberg et al., 2015) to predict a landmark position in each example. This is similar to the work of Esteves et al. (2018), which achieved exact equivariance to 2D rotation and scale and approximate equivariance to translation.

The first mention of Lorentz/Poincaré-equivariant networks was in Cheng et al. (2019), though they did not construct one. Concurrently to our own work, Bogatskiy et al. (2020) constructed a Lorentz-equivariant model which operated on irreps of the Lorentz group, derived similarly to appendix A.1. That work also made use of the Clebsch-Gordan coefficients and applied the model to experimental particle physics rather than object-tracking. It did not address how one might obtain irreps for other groups beyond the Lorentz group. Another line of work Finzi et al. (2020, 2021) concurrent to our own proposed a framework for building models equivariant to arbitrary Lie groups using exponential and logarithm maps between Lie algebra and group. However their models compute on tensor powers of a fixed representation and do not provide a technique for identifying the irreps. Our work complements this by providing an algorithm (FindRep) that solves for the irreps numerically. Practical advantages of using irrep features include lower dimensionality and better interpretability. For example, in the case of SO(3), 3- and 5- dimensional irrep features could be interpreted analogously to familiar geometric objects: vectors and Cauchy stress tensors, respectively.

## 2. Technical Background

We explain the most crucial concepts in this section. We defer an analytic derivation of the representation matrices of the Lorentz group to appendix A.1.

### 2.1. Symmetry Groups SO($n$) and SO($m, n$)

A 3D rotation may be defined as a matrix $A :\in \mathbb{R}^{3\times3}$ which satisfies the following properties, in which $\langle \boldsymbol{u}, \boldsymbol{v} \rangle = \sum_{i=1}^{3} u_i v_i$:

$$(\text{i}) \quad \det A = 1 \qquad (\text{ii}) \quad \forall \boldsymbol{u}, \boldsymbol{v} \in \mathbb{R}^3, \langle A\boldsymbol{u}, A\boldsymbol{v} \rangle = \langle \boldsymbol{u}, \boldsymbol{v} \rangle;$$

these imply the set of 3D rotations forms a group under matrix multiplication and this group is denoted SO(3). This definition directly generalizes to the $n-$dimensional rotation group SO($n$). For $n \geq 3$, the group SO($n$) is noncommutative, meaning there are elements $A, B \in$ SO($n$) such that $AB \neq BA$. Allowing for rotations and translations of $n$ dimensional space gives the $n-$dimensional special Euclidean group SE($n$).

SO($n$) is generalized by a family of groups denoted SO($m, n$), with SO($n$) = SO($n, 0$). For integers $m, n \geq 0$, we define $\langle \boldsymbol{u}, \boldsymbol{v} \rangle_{m,n} = \sum_{i=1}^{m} u_i v_i - \sum_{i=m+1}^{m+n} u_i v_i$. The group SO($m, n$)

is the set of matrices $A \in \mathbb{R}^{(m+n)\times(m+n)}$ satisfying (i-ii) below:

$$\text{(i)} \ \det A = 1 \qquad \text{(ii)} \ \forall \boldsymbol{u}, \boldsymbol{v} \in \mathbb{R}^3, \langle A\boldsymbol{u}, A\boldsymbol{v} \rangle_{m,n} = \langle \boldsymbol{u}, \boldsymbol{v} \rangle_{m,n};$$

these imply that $\mathrm{SO}(m,n)$ is also a group under matrix multiplication. While the matrices in $\mathrm{SO}(n)$ can be seen to form a compact manifold for any $n$, the elements of $\mathrm{SO}(m,n)$ form a noncompact manifold whenever $n, m \geq 1$. For this reason $\mathrm{SO}(n)$ and $\mathrm{SO}(m,n)$ are called compact and noncompact Lie groups respectively. The representations of compact Lie groups are fairly well-understood, see Bump (2004); Cartan (1930).

### 2.2. Action of $\mathrm{SO}(m,n)$ on Spacetime

We now explain the physical relevance of the groups $\mathrm{SO}(m,n)$ by reviewing *spacetime*. We refer to Feynman et al. (2011) (ch. 15) for a pedagogical overview. Two observers who are moving at different velocities will in general disagree on the coordinates $\{(t_i, \boldsymbol{u}_i)\} \subset \mathbb{R}^4$ of some events in spacetime. Newton and Galileo proposed that they could reconcile their coordinates by applying a spatial rotation and translation (i.e., an element of SE(3)), a temporal translation (synchronizing their clocks), and finally applying a transformation of the following form:

$$t_i \mapsto t_i \qquad \boldsymbol{u}_i \mapsto \boldsymbol{u}_i + \boldsymbol{v}t_i, \tag{1}$$

in which $\boldsymbol{v}$ is the relative velocity of the observers. The transformation (1) is called a *Galilean boost*. The set of all Galilean boosts along with 3D rotations forms the homogeneous Galilean group denoted $\mathrm{HG}(1,3)$. Einstein argued that (1) must be corrected by adding terms dependent on $||\boldsymbol{v}||_2/c$, in which $c$ is the speed of light and $||\boldsymbol{v}||_2$ is the $\ell_2$ norm of $\boldsymbol{v}$. The resulting coordinate transformation is called a *Lorentz boost*, and an example of its effect is shown in figure 1. The set of 3D rotations along with Lorentz boosts is exactly the group $\mathrm{SO}(3,1)$. In the case of 2 spatial dimensions, the group is $\mathrm{SO}(2,1)$. Including spacetime translations along with the Lorentz group $\mathrm{SO}(n,1)$ gives the larger Poincaré group $\mathcal{P}_n$ with $n$ spatial dimensions. The Poincaré group $\mathcal{P}_3$ is the group of coordinate transformations between different observers in special relativity.

Consider an object tracking task with input data consisting of a spacetime point cloud with $n$ dimensions of space and 1 of time, and corresponding outputs consisting of object class along with location and velocity vectors. A perfectly accurate object tracking model must respect the action of $\mathcal{P}_n$ on the input. That is, given the spacetime points in *any* observer's coordinate system, the perfect model must give the correct outputs *in that coordinate system*. Therefore the model should be $\mathcal{P}_n$-equivariant. For low velocities the symmetries of the homogeneous Galilean groups $\mathrm{HG}(n,1)$ provide a good approximation to $\mathrm{SO}(n,1)$ symmetries, so Galilean-equivariance may be sufficient for some tasks. Unfortunately the representations of $\mathrm{HG}(n,1)$ are not entirely understood although progress continues on this important problem (Levy-Leblond, 1971; De Montigny et al., 2006; Niederle and Nikitin, 2006).

### 2.3. Lie Groups and Lie Algebras

Here we give an intuitive summary of Lie groups and Lie algebras, deferring to Bump (2004) for a rigorous technical background. A Lie group $G$ gives rise to a *Lie algebra $A$* as

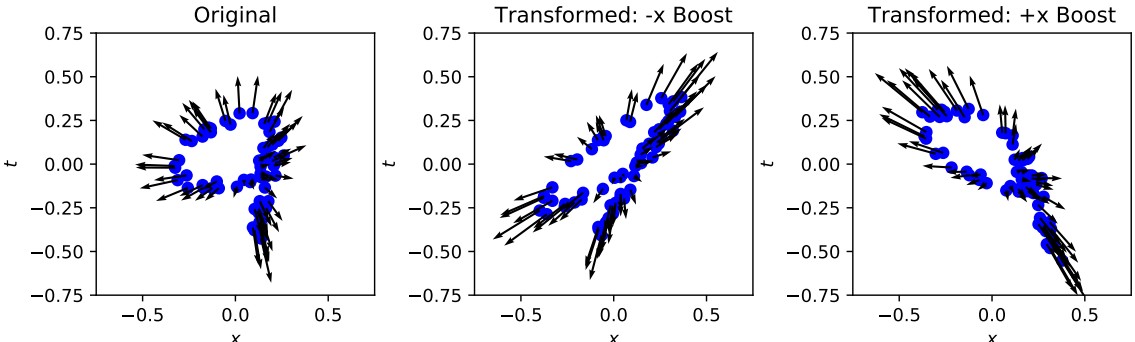

Figure 1: Activations of an $SO(2, 1)$-Equivariant neural network constructed using our framework. The arrows depict the elements of the 3-dimensional representation space (arrows) and are embedded on their associated points within the point cloud. This point cloud is from the MNIST-Live dataset as generated with digits embedded in the $x - t$ plane. The $y$ axis is suppressed. The left plot depicts the "original" activations (with the digit at rest). The right plots show what happens if we transform the point cloud with a Lorentz boost in the $\pm x$ direction before feeding it through the network. As dictated by Lorentz-equivariance, the activation vectors generated by the network transform in the same way as the input point cloud.

its *tangent space* at the identity. This is a vector space $V$ along with a bilinear product called the *Lie bracket*: $[a, b]$ which must behave like[2] the commutator for an associative ring $R$ with multiplication operation $\times_R$:

$$[a, b] = a \times_R b - b \times_R a$$

The Lie algebra for SO(3), denoted $\mathfrak{so}(3)$, has a basis $\{J_1, J_2, J_3\}$ satisfying

$$[J_i, J_j] = \epsilon_{ijk} J_k, \tag{2}$$

in which $\epsilon_{ijk} \in \{\pm 1, 0\}$ is the totally antisymmetric Levi-Civita symbol.[3] Intuitively, the Lie bracket shows how group elements near the identity fail to commute. For example, the matrices $R_x, R_y, R_z$ for rotations about the $x$ and $y$ axes by a small angle $\theta$ satisfy

$$R_x R_y - R_y R_x = R_z + O(\theta^2);$$

more generally the Lie bracket of (2) is satisfied to first order in $\theta$. The Lie algebra $\mathfrak{so}(3, 1)$ of the Lorentz Group SO(3, 1) also satisfies (2) for the generators $J_1, J_2, J_3$ of its subalgebra isomorphic to $\mathfrak{so}(3)$. It has 3 additional generators denoted $K_1, K_2, K_3$, which satisfy:

$$[J_i, K_j] = \epsilon_{ijk} K_k \qquad [K_i, K_j] = -\epsilon_{ijk} J_k \tag{3}$$

---

2. Specifically, the Lie bracket must satisfy the *Jacobi identity* and $[a, a] = 0$.

3. The symbol $\epsilon_{ijk}$ simply expresses in (2) that $[J_1, J_2] = J_3, [J_2, J_3] = J_1, [J_3, J_1] = J_2$.

These $K_i$ correspond to the Lorentz boosts in the same way that the $J_i$ correspond to the rotations. In general, if $\mathcal{A}$ is a $t$-dimensional Lie algebra with generators $T_1, ..., T_t$ such that

$$[T_i, T_j] = \sum_{k=1}^{t} A_{ijk} T_k, \tag{4}$$

we call the tensor $A_{ijk}$ the *structure constants* of $\mathcal{A}$.

## 2.4. Group Representations and the Tensor Product

Let $G$ be a Lie group and $\rho : G \to \mathbb{R}^{n \times n}$ be a represenation of $G$ as defined in section 1.1. Then $\rho$ defines a *group action* on $\mathbb{R}^n$: given a vector $\boldsymbol{u} \in \mathbb{R}^n$ and a group element $\alpha \in G$, we can define

$$\alpha *_\rho \boldsymbol{u} := \rho(\alpha)\boldsymbol{u}$$

using the matrix product. We then say that $\rho$ is *irreducible* if it leaves no nontrivial subspace invariant – for every subspace $V \subset \mathbb{R}^n$ with $0 < \dim V < n$, there exists $\alpha \in G, \boldsymbol{v} \in V$ such that $\alpha \star_\rho \boldsymbol{v} \notin V$.

Given two $G$-representations $\rho_1 : G \to \mathbb{R}^{n_1 \times n_1}$, $\rho_2 : G \to \mathbb{R}^{n_2 \times n_2}$, we define their *tensor product* as

$$\rho_1 \otimes \rho_2 : G \to \mathbb{R}^{n_1 n_2 \times n_1 n_2} \qquad (\rho_1 \otimes \rho_2)(\alpha) = \rho_1(\alpha) \otimes \rho_2(\alpha),$$

in which $\otimes$ on the right hand side denotes the usual tensor product of matrices. It is easy to check that $\rho_1 \otimes \rho_2$ is also a representation of $G$ using the fact that for matrices $A_1, A_2 \in \mathbb{R}^{n_1}$ and $B_1, B_2 \in \mathbb{R}^{n_2 \times n_2}$,

$$(A_1 \otimes B_1)(A_2 \otimes B_2) = (A_1 A_2) \otimes (B_1 B_2).$$

For $\rho_1, \rho_2$ as above we also define their *direct sum* as

$$(\rho_1 \oplus \rho_2)(\alpha) = \left( \begin{array}{c|c} \rho_1(\alpha) & \\ \hline & \rho_2(\alpha) \end{array} \right).$$

For two groups $H, G$ we say that $H$ is isomorphic to $G$ and write $H \cong G$ if there exists a bijection $f : H \to G$ such that $f(\alpha\beta) = f(\alpha)f(\beta)$. For $\rho_1, \rho_2$ as above, their images $\rho_i(G)$ are groups; we say that $\rho_1$ and $\rho_2$ are *isomorphic* and write $\rho_1 \cong \rho_2$ only if $\rho_1(G) \cong \rho_2(G)$. Some familiar representations of SO(3) act on scalars $\in \mathbb{R}$, vectors $\in \mathbb{R}^3$, and tensors (e.g., the Cauchy stress tensor) – these representations are all nonisomorphic. For many Lie groups such as SO$(n, 1)$ and SO$(n)$, a property called *complete reducibility* guarantees that any representation is either irreducible, or isomorphic to a direct sum of irreps. For such groups it suffices to identify the irreps to understand all other representations and construct equivariant models.

## 2.5. Clebsch-Gordan Coefficients and Tensor-Product Nonlinearities

**Clebsch-Gordan Coefficients:** Let $G$ be a completely reducible Lie group, and let $\rho_1, \rho_2, \rho_3$ be irreducible $G$-representations on the vector spaces $\mathbb{R}^{n_1}, \mathbb{R}^{n_2}, \mathbb{R}^{n_3}$. Consider the tensor product representation $\rho_1 \otimes \rho_2$. Since $G$ is completely reducible, there exists a

set $S$ of irreps such that $\rho_1 \otimes \rho_2 \cong \bigoplus_{\rho \in S} \rho$. Suppose that $\rho_3 \in S$. Then there exists a matrix $C \in \mathbb{R}^{n_3 \times (n_1 n_2)}$ which projects the space of the $n_3$-dimensional group representation $\rho_3$ from the tensor product space $\mathbb{R}^{n_1} \otimes \mathbb{R}^{n_2}$. That is,

$$\forall (\alpha, \boldsymbol{u}, \boldsymbol{v}) \in G \times \mathbb{R}^{n_1} \times \mathbb{R}^{n_2}, \; C(\rho_1(\alpha) \otimes \rho_2(\alpha))(\boldsymbol{u} \otimes \boldsymbol{v}) = \rho_3(\alpha) C(\boldsymbol{u} \otimes \boldsymbol{v})$$
$$\Rightarrow C(\rho_1(\alpha) \otimes \rho_2(\alpha)) = \rho_3(\alpha) C. \tag{5}$$

The matrices $C$ satisfying (5) for various $\rho_3$ are called the *Clebsch-Gordan coefficients*. In (5) there are $n_1 n_2 n_3$ linear constraints on C, and therefore this is a well-posed homogeneous linear program (LP) for $C$. The entries of $C$ may be found numerically by sampling several distinct $\alpha \in G$ and concatenating the linear constraints ((5)) to form the final LP. The solutions for $C$ form a linear subspace of $\mathbb{R}^{n_3 \times (n_1 n_2)}$ given by the nullspace of some matrix we denote $\mathcal{C}[\rho_1, \rho_2, \rho_3]$.

**Tensor Product Nonlinearities:** Tensor product nonlinearities, including norm non-linearities, use the Clebsch-Gordan coefficients defined above to compute equivariant quadratic functions of multiple $G$-representations within the $G$-equivariant model. This was demonstrated for the case of SE(3) by Thomas et al. (2018); Kondor et al. (2018) and for SO(3, 1) by Bogatskiy et al. (2020).

## 3. Methods

### 3.1. Learning Lie Group Representations

For a matrix $M \in \mathbb{R}^{n \times n}$ we denote its Frobenius and $L_1$ norms by

$$|M|_F^2 = \sum_{1 \le i,j \le n} |M_{ij}|^2, \qquad |M|_1 = \sum_{1 \le i,j \le n} |M_{ij}|.$$

FindRep first learns a Lie algebra representation and then obtains its corresponding group representation through the matrix exponential. For Lie groups that are *simply connected* as manifolds, all of their group representations are accessible through this approach; the groups we consider here are all simply connected. For other groups with multiple connected components, the full set of representations are characterized by additional discrete generators which are not produced by FindRep. See for example (Finzi et al., 2021) for further discussion of this case.

Fix a $t$-dimensional Lie algebra $\mathcal{A}$ with structure constants $A_{ijk}$ as defined in (4). Fix a positive integer $n$ as the dimension of the representation of $\mathcal{A}$. Let the matrices $T_1, ..., T_t \in \mathbb{R}^{n \times n}$ be optimization variables, and define the following loss function on the $T_i$:

$$\mathcal{L}[T_1, ..., T_t] = \underbrace{\max\left(1, \max_{1 \le i \le t} \frac{1}{|T_i|_F^2}\right)}_{N[T_i]^{-1} :=} \times \sum_{1 \le i \le j \le t} \left| [T_i, T_j] - \sum_k A_{ijk} T_k \right|_1. \tag{6}$$

This is the magnitude of violation of the structure constants of $\mathcal{A}$, times a norm penalty term $N[T_i]^{-1}$. The norm penalty prevents convergence to the trivial solution $(T_i)_{jk} = 0 \; \forall (i, j, k) \in [t] \times [n] \times [n]$. We pose the non-convex optimization problem $\min_{T_i \in \mathbb{R}^{n \times n}} \mathcal{L}[T_1, ..., T_t]$. We initialize the $T_i$ with entries from the standard normal distribution and perform gradient

descent in PyTorch with the adam optimizer (Kingma and Ba, 2014) with initial learning rate 0.1. The learning rate is set to decrease exponentially when loss plateaus. The results are shown in figure 2.

## 3.2. Verifying Irreducibility of Learned Representations

Suppose we have converged to $T_1, \ldots T_t$ such that $\mathcal{L}[T_i] = 0$. Then the $T_1, ..., T_t$ are a nonzero $n$-dimensional representation of the Lie *algebra* $\mathcal{A}$. The groups considered here are covered by the exponential map applied to their Lie algebras, so for each $\alpha \in G$ there exist $b_1, \ldots, b_t \in \mathbb{R}$ such that $\rho(\alpha) = \exp\left[\sum_{i=1}^{t} b_i T_i\right]$, where $\rho$ is any $n-$dimensional representation of $G$ and exp is the matrix exponential. This $\rho : G \mapsto \mathbb{R}^{n \times n}$ is then a representation of the Lie *group*. Throughout this section, $\rho$ denotes this representation. In general $\rho$ may leave some nontrivial subspace invariant. In this case it is *reducible* and splits as the direct sum of lower-dimensional irreps $\rho_i$ as explained in 2.4: $\rho \cong \rho_1 \oplus \ldots \oplus \rho_\ell$. Recall that any representation may be obtained as such a direct sum of irreps, so it is important to verify that $\rho$ is indeed irreducible, corresponding to $\ell = 1$. To validate that $\rho$ is irreducible, FindRep computes its tensor product structure and compares with the expected structure. Specifically, it computes the Clebsch-Gordan coefficients for the direct-sum decomposition of the tensor product of the learned representation $\rho$ with several other known representations $\rho_1, ..., \rho_r$. section 2.5 defines these coefficients and explains how they are computed from the nullspace of the matrix $\mathcal{C} = \mathcal{C}[\rho, \rho_1, \rho_2]$, in which $\rho_2$ appears in the decomposition of $\rho \otimes \rho_1$. Let $\rho_1, \rho_2$ denote two other known representations, and consider the Clebsch-Gordan coefficients $C$ such that $C\rho \otimes \rho_1 = \rho_2 C$. The dimension of the nullspace of $\mathcal{C}$ indicates the number of unique nonzero matrices $C$ of Clebsch-Gordan coefficients. The singular values of $\mathcal{C}$ are denoted $SV_1(\mathcal{C}) \leq ... \leq SV_\ell(\mathcal{C})$. The ratio

$$r(\mathcal{C}) := SV_2(\mathcal{C})/SV_1(\mathcal{C}) \tag{7}$$

diverges only if the nullspace is one dimensional which therefore corresponds to a unique solution for $C$. The number of expected solutions is known (e.g., it may be computed using the same technique from the formulae for the irreps). Therefore if $r(\mathcal{C})$ diverges for exactly the choices of $\rho_1, \rho_2$ where the theory indicates that unique nonzero Clebsch-Gordan coefficients exist, then this is consistent with our having learned an irrep of the group $G$. Further, when $\rho_1$ is the trivial representation (i.e. $\rho_1(\alpha) = 1 \forall \alpha$), we clearly have $\rho \otimes \rho_1 = \rho$. In this case, the permissible $C$ correspond to $G-$linear maps $\mathbb{R}^n \to \mathbb{R}^{n_2}$. By a result of Schur (1905) (Schur's Lemma), the only such (nonzero) maps are isomorphisms. Therefore a divergent value of $r(\mathcal{C})$ when $\rho_1 = 1$ indicates that $\rho \cong \rho_2$. This is shown in the top row of figure 3 and discussed in section 4.

Similar to (Rao and Ruderman, 1999), FindRep restarts gradient descent starting from random initialization points. A restart is triggered if loss plateaus and the learning rate is smaller than the loss by a factor of at most $10^{-4}$. The tensor product structure is computed upon convergence to loss under $10^{-9}$, and a restart is triggered if the divergences of $r(\mathcal{C})$ do not agree with the theoretical prediction, indicating a reducible representation.

## 4. Experiments

Code to reproduce all experiments is available online[4]. We apply FindRep to $SO(3), SO(2, 1)$, and $SO(3, 1)$ to find (respectively) $3, 3$, and $4$ dimensional irreps. Restarts due to loss plateaus or convergence to reducible representations are common, occurring 0, 19, and 17 times, respectively. After the restarts, the loss converges arbitrarily close to 0 with the penalty term bounded above by a constant. We exponentiate the resulting algebra representation matrices to obtain group representations and calculate the tensor product structure as described in section 3.2. The details of this calculation are in appendix A.6 and shown in figure 3. The results indicate that the learned representations are irreps of the associated Lie algebras to within numerical error of about $10^{-6}$. Schur's Lemma in the special case of the tensor product with the trivial representation indicates the isomorphism class of each learned group representation.

To illustrate the utility of these learned irreps, we constructed a Poincaré-equivariant neural network architecture called SpacetimeNet and applied it to a relativistic object-tracking task. These results are described in the appendix.

### 4.1. Conclusion

We envision many applications of Poincaré-equivariant deep neural networks beyond the physics of particles and plasmas. SpacetimeNet can identify and track simple objects as they move through 3D space. This suggests that Lorentz-equivariance is a useful prior for object-tracking tasks. With a treatment of bandlimiting and resampling as in Worrall et al. (2017); Weiler et al. (2018), our work could be extended to build Poincaré-equivariant networks for volumetric data. More broadly, understanding the representations of noncompact and noncommutative Lie groups may enable the construction of networks equivariant to new sets of symmetries such as the Galilean group. Since the representation theory of these groups is not entirely understood, automated techniques such as FindRep could be beneficial.

## Acknowledgments

We thank Ryan Loney and Rosario Cammarota for computing support and numerous helpful discussions. We also thank anonymous reviewers for helpful comments and suggestions.

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

## Appendix A. Appendix

### A.1. Analytic Derivation of Lorentz Group Representations

To compare our learned group representations with those obtained through prior methods, we require analytical formulae for the Lie algebra representations for the algebras $\mathfrak{so}(3), \mathfrak{so}(3,1)$, and $\mathfrak{so}(2,1)$. The case of $\mathfrak{so}(3)$ has a well-known solution (see Griffiths and Griffiths (2005)). If complex matrices are permissible the library QuTiP Johansson et al.

(2013) has a function "jmat" that readily gives the representation matrices. A formulae to obtain real-valued representation matrices is given in Pinchon and Hoggan (2007) and a software implementation is available at Cohen et al. (2020). The three-dimensional Lie algebra $\mathfrak{so}(2,1) = \text{span}\{K_x, K_y, J_z\}$ has structure constants given by (3). In fact, these three generators $K_x, K_y, J_z$ may be rescaled so that they satisfy (2) instead. This is due to the isomorphism $\mathfrak{so}(3) \cong \mathfrak{so}(2,1)$. Specifically, leting $\{L_x, L_y, L_z\}$ denote a Lie algebra representation of $\mathfrak{so}(3)$, defining

$$K_x = -iL_x \qquad K_y = -iL_y \qquad J_z := L_z,$$

it may be easily checked that $K_x, K_y, J_z$ satisfy the applicable commutation relations from Equation (3). This reflects the physical intuition that time behaves like an imaginary dimension of space.

The final Lie algebra for which we require explicit representation matrix formulas is $\mathfrak{so}(3,1)$. Following Weinberg (1995), we define new generators $A_i, B_i$ as

$$A_i := \frac{1}{2}(J_i + iK_i) \qquad B_i := \frac{1}{2}(J_i - iK_i), \tag{8}$$

we see that the $\mathfrak{so}(3,1)$ commutators (2), (3) become

$$[A_i, A_j] = i\epsilon_{ijk}A_k, \qquad [B_i, B_j] = i\epsilon_{ijk}B_k, \qquad [A_i, B_j] = 0. \tag{9}$$

Therefore $\mathfrak{so}(3,1) \cong \mathfrak{so}(3) \oplus \mathfrak{so}(3)$, and the irreducible algebra representations of $\mathfrak{so}(3,1)$ may be obtained as the direct sum of two irreducible algebra representations of $\mathfrak{so}(3)$.

## A.2. SpacetimeNet Architecture

We obtain all Clebsch-Gordan coefficients through the procedure explained in section 2.5. We place them in a tensor: $C_{g,qr,ls,mt}$. This notation corresponds to taking the tensor product of an element of the $l^{\text{th}}$ group representation space indexed by $s$ with an element of the $m^{\text{th}}$ group representation space indexed by $t$, and projecting it onto the $q^{\text{th}}$ group representation space indexed by $r$. The space of possible Clebsch-Gordan coefficients can be multidimensional.[5] We use an index $g$ to carry the dimension within the space of Clebsch-Gordan coefficients.

The trainable weights in SpacetimeNet are complex-valued filter weights denoted $f_{qg}^k$ and channel-mixing weights denoted $W_{qcgd}^k$. Each layer builds a collection of equivariant convolutional filters $F_{xijqr}^k$ from the geometry of the point cloud. Let $q'$ denote the index of the group representation in which the points are embedded. Let $X_{xir}$ denote the point coordinates, in which $x$ indexes the batch dimension, $i$ indexes the points, and $r$ indexes the $q'$ group representation space. Define the (globally) translation-invariant quantity $\Delta X_{xijr} := X_{xjr} - X_{xir}$. The equivariant filters at layer $k$ are:

$$F_{xijqr}^k = \delta_{qq'}\Delta X_{xijr} + \sum_{s,t,g} C_{g,qr,q's,q't} f_{qg}^k \Delta X_{xijs}\Delta X_{xijt}. \tag{10}$$

---

5. This is common if a group representation is itself obtained via tensor product.

The input and activations for the $k^{\text{th}}$ layer of the network are defined on a tensor $V^k_{ximct}$, where $x$ is the batch dimension, $i$ indexes the points, $m$ is the group representation index, $c$ is the channel index, $t$ indexes the group representation space. Our mixing weights are then defined for the $k^{\text{th}}$ layer as $W^k_{qcgd}$ with layer update rule:

$$V^{k+1}_{xiqcr} = \sum_{g,l,s,m,t,d,j} C_{g,qr,ls,mt} F^k_{xijls} V^k_{xjmdt} W^k_{qcgd}. \tag{11}$$

A proof that SpacetimeNet is $\mathcal{P}_n$-equivariant is given in appendix A.4.

### A.3. Poincaré-Equivariant Object-Tracking Networks

We created MNIST-Live, a benchmark dataset of spacetime point clouds sampled from digits in the MNIST dataset moving uniformly through space. Each sample consists of 64 points with uniformly random times $t \in [-1/2, 1/2]$, and spatial coordinates sampled from a 2D probability density function proportional to the pixel intensity. Using instances of the 0 and 9 classes, we train on examples with zero velocity and a fixed orientation (i.e., a fixed reference frame) and evaluate on examples with random velocity and orientation. This dataset is analogous to data from an event camera (see (Orchard et al., 2015)) or LIDAR system. We train 3 layer $SO(2, 1)$ and $SO(3, 1)$-equivariant SpacetimeNet models with 3 channels and batch size 16 on 4096 MNIST-Live examples and evaluate on a development set of 124 examples. We obtain an accuracy of $80 \pm 5\%$ as shown in figure 4. Note conventional perceptrons or CNN are known to obtain significantly higher accuracy by taking advantage of a fixed reference frame. However, SpacetimeNet's accuracy is invariant under the choice of reference frame, up to machine precision, since SpacetimeNet is fully Poincaré-equivariant as proved in appendix A.4. This allows SpacetimeNet to generalize from training samples with a fixed reference frame to test samples with random reference frames. A conventional perceptron or CNN model exhibits no such guarantee and the accuracy will generally degrade if the reference frame of the input data is altered.

### A.4. Proof that SpacetimeNet is Poincaré-Equivariant

Consider an arbitrary Poincaré group transformation $\alpha \in \mathcal{P}_n$, and write $\alpha = \beta t$ in which $\beta \in SO(n, 1)$ and $t$ is a translation. Suppose we apply this $\alpha$ to the inputs of (10) through the representations indexed by $q$: $\rho_q(\alpha)_{st}$, in which $s, t$ index the representation matrices.

Then since the translation $t$ leaves $\Delta X$ invariant, the resulting filters will be

$$
\begin{aligned}
F^k_{xijqr} &= \delta_{qq'} \sum_{r'} \rho_{q'}(\beta)_{rr'} \Delta X_{xijr'} + \sum_{s,t,g} C_{g,qr,q's,q't} f^k_{qg} \sum_{s',t'} \rho_{q'}(\beta)_{ss'} \Delta X_{xijs'} \rho_{q'}(\beta)_{tt'} \Delta X_{xijt'} \\
&= \delta_{qq'} \sum_{r'} \rho_{q'}(\beta)_{rr'} \Delta X_{xijr'} + \sum_{g,s',t'} \left( \sum_{s,t} C_{g,qr,q's,q't} \rho_{q'}(\beta)_{ss'} \rho_{q'}(\beta)_{tt'} \right) f^k_{qg} \Delta X_{xijs'} \Delta X_{xijt'} \\
&= \delta_{qq'} \sum_{r'} \rho_{q'}(\beta)_{rr'} \Delta X_{xijr'} + \sum_{s,t,g,r'} \left( \rho_q(\beta)_{rr'} C_{g,qr',q's,q't} \right) f^k_{qg} \Delta X_{xijs} \Delta X_{xijt} \\
&= \sum_{r'} \rho_{q'}(\beta)_{rr'} \left( \delta_{qq'} \Delta X_{xijr'} + \sum_{s,t,g,r'} C_{g,qr',q's,q't} f^k_{qg} \Delta X_{xijs} \Delta X_{xijt} \right) \\
&= \sum_{r'} \rho_{q'}(\beta)_{rr'} F^k_{xijqr'},
\end{aligned}
$$

where we have used (5). The network will be equivariant if each layer update is equivariant. Recall the layer update rule of (11):

$$
V^{k+1}_{xiqcr} = \sum_{g,l,s,m,t,d,j} C_{g,qr,ls,mt} F^k_{xijls} V^k_{xjmdt} W^k_{qcgd}.
$$

Suppose for the same transformation $\alpha = \beta t$ above, that $V^k$ and $\Delta X$ are transformed by $\alpha$. Then because the activations associated with each point are representations of $\mathrm{SO}(n,1)$, they are invariant to the global translation $t$ of the point cloud and we have

$$
\begin{aligned}
V^{k+1}_{xiqcr} &= \sum_{g,l,s,m,t,d,j} C_{g,qr,ls,mt} \sum_{s'} \rho_m(\beta)_{ss'} F^k_{xijls'} \sum_{t'} \rho_m(\beta)_{tt'} V^k_{xjmdt'} W^k_{qcgd} \\
&= \sum_{s',t'} \sum_{g,l,s,m,t,d,j} \left( C_{g,qr,ls,mt} \rho_m(\beta)_{ss'} \rho_m(\beta)_{tt'} \right) F^k_{xijls'} V^k_{xjmdt'} W^k_{qcgd} \\
&= \sum_{g,l,s,m,t,d,j,r'} \left( \rho_m(\beta)_{rr'} C_{g,qr',ls,mt} \right) F^k_{xijls} V^k_{xjmdt} W^k_{qcgd} \\
&= \sum_{r'} \rho_m(\beta)_{rr'} V^{k+1}_{xiqcr'},
\end{aligned}
$$

where again we applied (5).

## A.5. Equivariant Convolutions

Consider data on a point cloud consisting of a finite set of spacetime points $\{\boldsymbol{x}_i\} \subset \mathbb{R}^4$, a representation $\rho_0 : \mathrm{SO}(3,1) \to \mathbb{R}^{4 \times 4}$ of the Lorentz group defining its action upon the spacetime, and feature maps $\{\boldsymbol{u}_i\}, \{\boldsymbol{v}_i\} \subset \mathbb{R}^n$ associated with representations $\rho_u : \mathrm{SO}(3,1) \to \mathbb{R}^{m \times m}$ and $\rho_v : \mathrm{SO}(3,1) \to \mathbb{R}^{n \times n}$. A convolution of this feature map can be written as

$$
\boldsymbol{u}'_i = \sum_j \kappa(\boldsymbol{x}_j - \boldsymbol{x}_i) \boldsymbol{u}_j
$$

in which $\kappa(\boldsymbol{x}) : \mathbb{R}^4 \to \mathbb{R}^{n \times m}$, a matrix-valued function of spacetime, is the filter kernel.

$\mathcal{P}_3$-equivariance dictates that for any $\alpha \in \mathrm{SO}(3,1)$,

$$\rho_v(\alpha) \sum_j \kappa(\boldsymbol{x}_j - \boldsymbol{x}_i)\boldsymbol{u}_j = \sum_j \kappa(\rho_1(\alpha)(\boldsymbol{x}_j - \boldsymbol{x}_i))\rho_u(\alpha)\boldsymbol{u}_j$$

$$\Rightarrow \kappa(\Delta\boldsymbol{x}) = \rho_v(\alpha^{-1})\kappa(\rho_0(\alpha)\Delta\boldsymbol{x})\rho_u(\alpha) \quad (12)$$

Therefore a single kernel matrix in $\mathbb{R}^{n \times m}$ may be learned for each coset of spacetime under the action of $\mathrm{SO}(3,1)$. The cosets are indexed by the invariant

$$t^2 - x^2 - y^2 - z^2.$$

The kernel may then be obtained at an arbitrary point $\boldsymbol{x} \in \mathbb{R}^4$ from (12) by computing an $\alpha$ that relates it to the coset representative $\boldsymbol{x}_0$: $\boldsymbol{x} = \rho_0(\alpha)\boldsymbol{x}_0$. A natural choice of coset representatives for $\mathrm{SO}(3,1)$ acting upon $\mathbb{R}^4$ is the set of points $\{(t,0,0,0) : t \in \mathbb{R}^+\} \cup \{(0,x,0,0) : x \in \mathbb{R}^+\} \cup \{(t,ct,0,0) : t \in \mathbb{R}^+\}$.

## A.6. Tensor Product Structure of Learned $\mathrm{SO}(3), \mathrm{SO}(2,1), \mathrm{SO}(3,1)$ Group Representations

We quantify the uniqueness of each set of Clebsch-Gordan coefficients using the diagnostic ratio $r(\mathcal{C})$ defined in eq. (7). Recall that the value of $r$ becomes large only if there is a nondegenerate nullspace corresponding to a unique set of Clebsch- For $\mathrm{SO}(3)$ and $\mathrm{SO}(2,1)$, the irreducible group representations are labeled by an integer which is sometimes called the *spin*. We label learned group representations with a primed $(i')$ integer. For the case of $\mathrm{SO}(3,1)$ the irreducible group representations are obtained from two irreducible group representations of $\mathfrak{so}(3)$ as explained in section A.1 and we label these representations with both spins i.e. $(s_1, s_2)$. We again label the learned group representations of $\mathrm{SO}(3,1)$ with primed spins, i.e. $(s_1', s_2')$. The tensor product structures of the representations is shown in figure 3.

We have produced a software library titled *Lie Algebraic Networks* (LAN) built on PyTorch, which derives all Clebsch-Gordan coefficients and computes the forward pass of Lie group equivariant neural networks. LAN also deals with Lie algebra representations, allowing for operations such as taking the tensor product of mutliple group representations. figure 5 demonstrates the LAN library. Starting from several representations for a Lie algebra, LAN can automatically construct a neural network equivariant to the associated Lie group with the desired number of layers and channels. We present our experimental results training $\mathrm{SO}(2,1)$ and $\mathrm{SO}(3,1)$-equivariant object-tracking networks in section A.3.

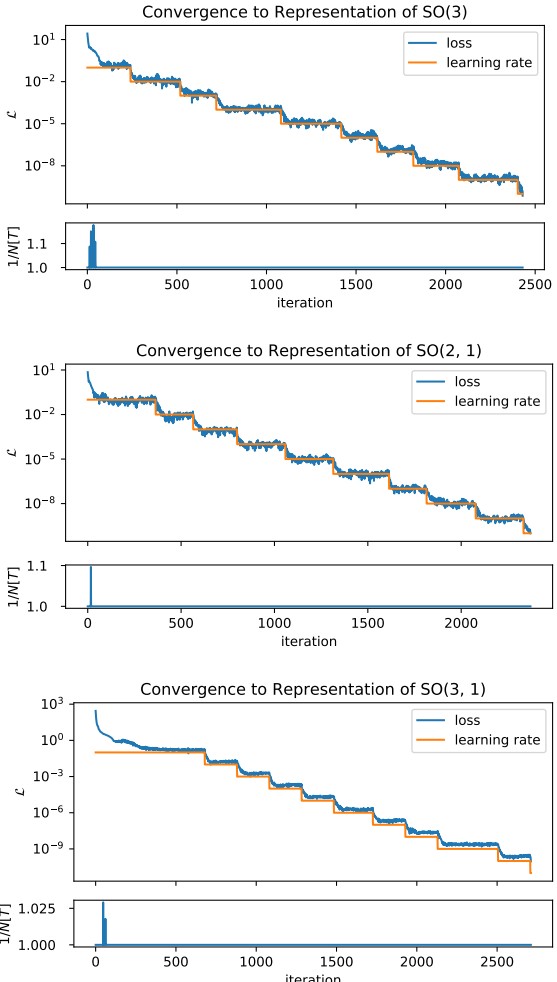

Figure 2: Convergence to arbitrary precision group representations of three Lie groups: SO(3), SO(2, 1), and SO(3, 1). The multiplicative norm penalty is plotted in each lower subplot, and demonstrates that this penalty is important early on in preventing the learning of a trivial representation, but for later iterations stays at its clipped value of 1. Loss is plotted on each upper subplot.

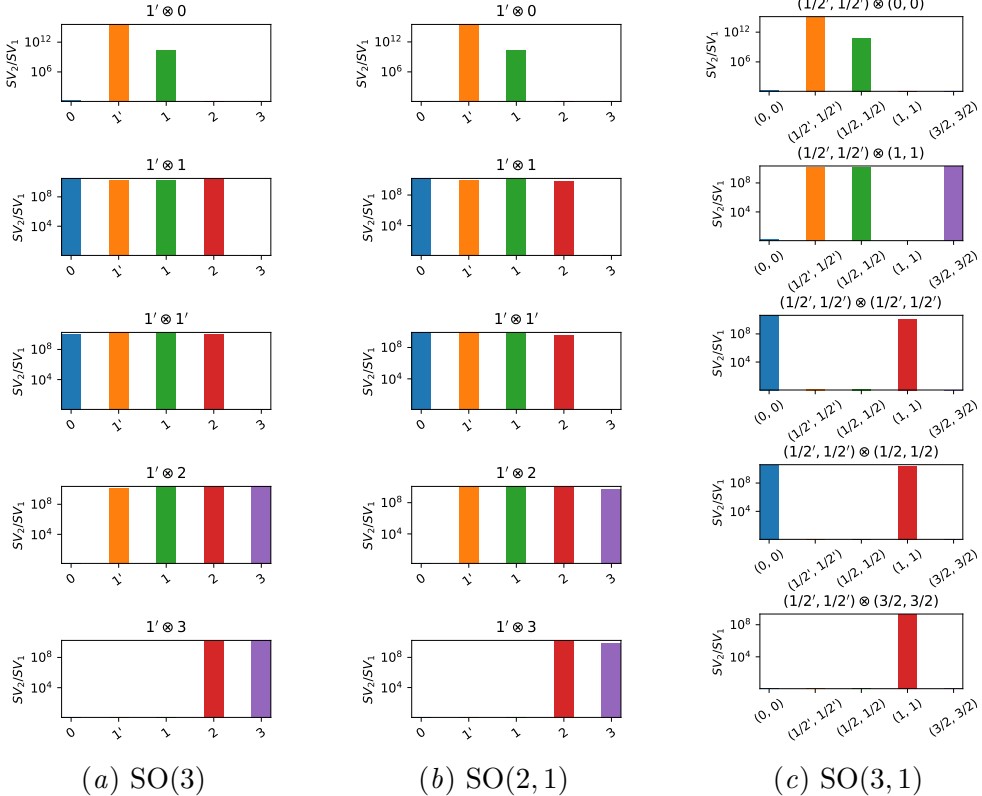

Figure 3: Tensor product structure of the learned group representations $\rho$ with several known (analytically-derived) group representations $\rho_1$ for the groups $\text{SO}(3), \text{SO}(2,1)$, and $\text{SO}(3,1)$. Each column is for the group indicated at the bottom, each row is for a different choice of $\rho_1$ for that group, and the horizontal axis indicates the $\rho^{(i)}$ onto which we project the tensor product $\rho \otimes \rho_1 \cong \oplus_{i \in I} \rho^{(i)}$. The diagnostic $r$ (defined by eq. (7) in section 2.5) is plotted on the $y$-axis with a log scale for each subfigure. The labelling of group representations is explained in section 4, recall that the primed integers indicate learned representations. The first row demonstrates by Schur's Lemma that to within numerical error of about $\sim 10^{-6}$ the learned $\text{SO}(3)$ group representation denoted $1'$ is isomorphic to the spin-1 irreducible group representation obtained from known formulae, i.e. $1'_{\text{SO}(3)} \cong 1_{\text{SO}(3)}$. The first row also indicates that $1'_{\text{SO}(2,1)} \cong 1_{\text{SO}(2,1)}$, and $(1/2', 1/2')_{\text{SO}(3,1)} \cong (1/2, 1/2)_{\text{SO}(3,1)}$. The remaining rows indicate that the tensor product structure of the learned group representations matches that of the known irreducible group representations.

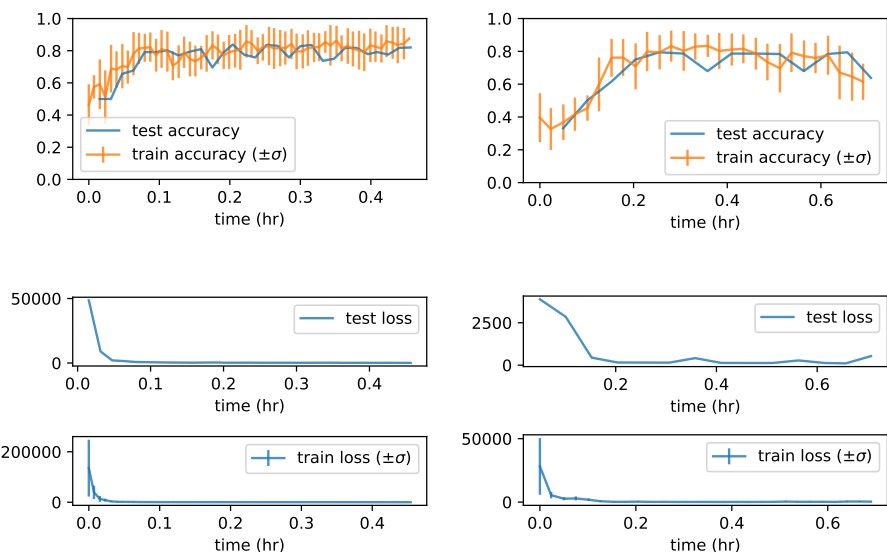

Figure 4: (Left) SO(2, 1)-equivariant neural network learning to recognize digits from the MNIST-Live dataset in 2 spatial dimensions. Error bars for train accuracy and loss are computed as the mean and standard deviation across a sliding window of 15 batches. (Right) SO(3, 1)-equivariant neural network training to recognize digits from the MNIST-Live dataset in 3 spatial dimensions. Error bars for train accuracy and loss are computed as the mean and standard deviation across a sliding window of 15 batches.

```python
1  from lan import LieAlgebraRepresentation, \
2      LieAlgebraRepresentationDirectSum, \
3      LieAlgebraTensorProductRepresentation, \
4      LieGroupEquivariantNeuralNetwork
5
6  learned_generators = [...]
7  known_generators = [...]
8
9  learned_irrep = LieAlgebraRepresentation(learned_generators)
10 scalar_irrep = LieAlgebraRepresentation(
11     numpy.zeros((
12         learned_irrep.algebra.dim, 1, 1
13     ))
14 )
15 known_irrep = LieAlgebraRepresentation(known_generators)
16
17 representations = LieAlgebraRepresentationDirectSum([
18     scalar_irrep,
19     known_irrep
20     learned_irrep,
21     LieAlgebraTensorProductRepresentation(
22         [learned_irrep, learned_irrep])
23 ])
24
25 model = LieGroupEquivariantNeuralNetwork(
26     representations, num_layers=10, num_channels=32)
```

Figure 5: Our Lie Algebraic Networks (lan) module handles Lie algebra and Lie group representations, derives Clebsch-Gordan coefficients for the equivariant layer update, and computes the forward pass. This makes it simple to build an equivariant point cloud network with the found representations. This software is available at github.com/noajshu/learning_irreps.

