# OpenReview forum: "Computing Representations for Lie Algebraic Networks"
_NeurIPS.cc/2022/Workshop/NeurReps — NeurReps 2022 Oral_

### Official Review · Reviewer_CjWV · 2022-10-04
**A valuable step forward for group-equivariant learning.**

**Confidence:** 4
**Soundness:** 3
**Presentation:** 3
**Contribution:** 3
**Overall Rating:** 7

**Summary:**

The paper introduces an algorithm for computing irreducible representations of arbitrary non-compact and non-commutative Lie groups. The proposed algorithm relies upon differentiable optimization, and only requires the structure constants of the associated Lie algebra to be given as input. As a second contribution, the paper goes further to the construction of Lie group-equivariant networks using the learned representations. These building blocks are showcased on classification problems over dynamic point clouds and equivariant object tracking, where the equivariance is understood with respect to the Poincare group.

**Questions:**

It would be interesting to see if irreducible representations can also be found by the proposed algorithm for higher-dimensional cases, even just from a numerical perspective without a practical grounding on real data. A possible suggestion, perhaps for future work: object tracking of high-dimensional point clouds arises in shape analysis and geometry processing, where a sequence of 3D shapes is represented using so-called spectral embeddings, i.e., each surface point is represented using the values at that point of the Laplacian eigenfunctions on the surface.

**Limitations:**

There is no discussion of possible failure cases of the multi-start optimization procedure. A brief discussion of cases where the procedure did not converge to an irreducible representation would be useful. Likewise, if no such cases ever occur, a discussion of this happening in practice would benefit the paper.

**Recommended Decision:**

3: Accept

**Relevance:**

4: Highly relevant

**Strengths And Weaknesses:**

As its main contribution, the paper introduces a novel solver for finding explicit representations (i.e. matrices with high numerical accuracy) for elements of non-compact and non-commutative Lie groups. This contribution is fundamental for the field of equivariant machine learning, with a good potential impact that could lead to further developments. See below for more details on the strengths (+) and weaknesses (-) of the paper:

(+) As stated above, the core contribution of this paper is rather fundamental in my opinion. There is a long line of works exploring the identification or the explicit construction of Lie group representations, but these either assume additional knowledge or are restricted to relatively simpler cases (e.g. compact groups). The paper provides an important additional step in this area, and could prove essential to move group-equivariant learning forward.

(+) The paper touches upon a complex topic while maintaining a good level of readability and accessibility for a general ML readership. Intuitive explanations are given throughout the main text, where possible. I think the paper is as as clear as it can get without being completely self-contained, which can be especially hard in this area.

(+) The experimental section includes real-world experiments on practical computer vision problems.

(-) A couple of recent works, mentioned in the introduction, might detract from the overall novelty of the method. The most relevant work was published in 2020 (although this submission still cites it as an arxiv preprint). The key difference to this work (other than the application of particle physics as opposed to object tracking) should be better described.

(-) The introduction also mentions that one of the practical advantages of irreducible representation features is their better interpretability, but does not further discuss this aspect. In what sense are these feature better interpretable?

(-) I think the paper organization might be improved by moving the more mathematical derivations to the Appendix, and moving sections A.3 and A.7 back to the main text. For instance, Figure 2 could fit the main paper by converting it to a single row.

A few minor comments:

- Section 2.1, equation (i): shouldn't it be $\det A = 1$?
- Can you please comment on the convexity of the energy in equation 6?
- In some cases, it would be better to use the \citep comment instead of \cite when citing the literature; for example, the last line of section 2.2 would better work with parentheses, while the last line of page 7 is good without parentheses.
- Typo on line 1 of page 6: "Lia algebra".
- Typo on page 9: "(i.e. ..." there is no closing parenthesis.
- The terminology used in Figure 2, namely "loss" and "learning rate" suggest there is a training process in action, when this is actually pure optimization. It would be better to use the classical optimization terms in this case.

**Submission Track:**

Proceedings Paper (9 Page)

---

### Official Review · Reviewer_UsgF · 2022-10-14
**A comuputational approach to the important but nontrivial problem of computing the representations of a symmetry group to build equivariant networks**

**Confidence:** 3
**Soundness:** 2
**Presentation:** 4
**Contribution:** 4
**Overall Rating:** 7

**Summary:**

The paper proposes an algorithm to compute numerically the representations of a group. This enables the construction of group equivariant neural networks when no explicit computation is known for the representation of the group. The method is validated on examples where we already have an exact computation given by an explicit formula. Finally the method is illustrated on an relativistic object tracking problem leveraging on the Poincare group equivariance, which encodes space-time transformations and echoes in many other applications including particle physics.

**Questions:**

One question. Where can we find the LAN library online ?

A small remark. Paragraph 3.11, if I am not mistaking, the method leverages on the one-to-one correspondence between the representations of a Lie a group and the ones of its Lie algebras which holds a priori only if the groupe $G$ is a simply connected Lie group. Maybe this point should be precised and a bit more detailed.

Some typos.
Paragraph 2.1. det(A)=1 not 0

Paragraph 2.2. I believe there may be a missing word or punctuation between the last sentence and the references.

Paragraph 2.3. Page 6, the Lie (not Lia) algebra

Paragraph 3.1.1. The title format. And $\rho_2$ rather than $\rho_3$ at the end of the page $8$.

Figure 3, where is diagnostic $r$ previously defined ?


**Limitations:**

The method is promising. To pursue its numerical and theoretical analysis, it would be interesting to study further the convergence of the method (convergence speed, computational cost, convergence guarantees...)

**Recommended Decision:**

3: Accept

**Relevance:**

4: Highly relevant

**Strengths And Weaknesses:**

The paper is well written and tackles an important problem as equivariant neural networks are gaining popularity. It provides with an explicit algorithm to implement the method, theoretical background, and several numerical validations. Additionally, it is illustrated on a meaningful application solving an object matching problem.

**Submission Track:**

Proceedings Paper (9 Page)

---

### Official Review · Reviewer_JWKH · 2022-10-16
**An interesting way to derive group representation from structure constants**

**Confidence:** 3
**Soundness:** 4
**Presentation:** 4
**Contribution:** 4
**Overall Rating:** 8

**Summary:**

The paper proposes a novel and interesting method for deriving the group representation corresponding to a given set of structure constants.  It is a nice step toward making the Lie group approach adaptive to different situations or applications, I believe it will be of interest to participants of the workshop.


**Questions:**

page 6, Lia algebra -->  Lie algebra


**Limitations:**

yes


**Recommended Decision:**

3: Accept

**Relevance:**

4: Highly relevant

**Strengths And Weaknesses:**

Strength:  a nice theoretical exposition and solution to a challenging problem
Weakness:  the paper may be inaccessible to those without strong background in Lie theory, but at the same time it addresses a highly technical issue and so I'm not sure this is avoidable.



**Submission Track:**

Proceedings Paper (9 Page)

---

### Decision · Program_Chairs · 2022-10-21

Accept (Oral)